# A novel microfluidic compact disc to investigate electrochemical property changes between artificial and real salivary samples mixed with mouthwashes using electrical impedance analysis

Aung Thiha[1,2], Fatimah Ibrahim[1,2,3,4,5]*, Karunan Joseph[1,2], Bojan Petrović[6]*, Sanja Kojić[7], Nuraina Anisa Dahlan[1,2], Nurul Fauzani Jamaluddin[1,2], Saima Qureshi[5], Goran M. Stojanović[7]

1 Department of Biomedical Engineering, Faculty of Engineering, Universiti Malaya, Kuala Lumpur, Malaysia, 2 Department of Biomedical Engineering, Centre for Innovation in Medical Engineering (CIME), Faculty of Engineering, Universiti Malaya, Kuala Lumpur, Malaysia, 3 Centre for Printable Electronics, Universiti Malaya, Kuala Lumpur, Malaysia, 4 Microwave Research Institute, Universiti Teknologi MARA, Shah Alam, Selangor, Malaysia, 5 Malaysian Research Institute on Ageing, Universiti Putra Malaysia, Serdang, Selangor, Malaysia, 6 Department of Dental Medicine, Faculty of Medicine, University of Novi Sad, Novi Sad, Serbia, 7 Faculty of Technical Science, University of Novi Sad, Novi Sad, Serbia

* fatimah@um.edu.my (FI); bojan.petrovic@mf.uns.ac.rs (BP)

## Abstract

Diagnosing oral diseases at an early stage may lead to better preventive treatments, thus reducing treatment burden and costs. This paper introduces a systematic design of a micro-fluidic compact disc (CD) consisting of six unique chambers that run simultaneously from sample loading, holding, mixing and analysis. In this study, the electrochemical property changes between real saliva and artificial saliva mixed with three different types of mouthwashes (i.e. chlorhexidine-, fluoride- and essential oil (Listerine)-based mouthwashes) were investigated using electrical impedance analysis. Given the diversity and complexity of patient's salivary samples, we investigated the electrochemical impedance property of healthy real saliva mixed with different types of mouthwashes to understand the different electrochemical property which could be a foundation for diagnosis and monitoring of oral diseases. On the other hand, electrochemical impedance property of artificial saliva, a commonly used moisturizing agent and lubricant for the treatment of xerostomia or dry mouth syndrome was also studied. The findings indicate that artificial saliva and fluoride-based mouthwash showed higher conductance values compared to real saliva and two other different types of mouthwashes. The ability of our new microfluidic CD platform to perform multiplex processes and detection of electrochemical property of different types of saliva and mouthwashes is a fundamental concept for future research on salivary theranostics using point-of-care microfluidic CD platform.

**Data Availability Statement:** All relevant data are within the manuscript and its Supporting information files.

**Funding:** This project has received funding from Universiti Malaya Partnership Grant RK006-2021, the European Union's Horizon 2020 research and innovation programme under the Marie Skłodowska-Curie grant agreement No. 872370 and 854194, and Universiti Malaya Impact Oriented Interdisciplinary Research Grant IIRG002A-2022HWB (WP1). The recipients of respective grants are as follows: Bojan Petrović, Sanja Kojić, Fatimah Ibrahim , Karunan Joseph are the recipients of the European Union's Horizon 2020 research and innovation programme under the Marie Skłodowska-Curie grant agreement No. 872370; Goran M. Stojanović and Saima Qureshi grant no. 854194. Fatimah Ibrahim , Aung Thiha, Nurul Fauzani Jamaluddin are the recipients of Universiti Malaya Partnership Grant (RK006-2021) and Universiti Malaya Impact Oriented Interdisciplinary Research Grant IIRG002A-2022HWB (WP1).

**Competing interests:** The authors have declared that no competing interests exist.

# Introduction

On-site testing of patients' salivary specimens could open access to rapid screening, diagnosis, and prognosis of oral diseases. Oral diseases are one of the most expensive medical conditions in Europe, behind diabetes and cardiovascular diseases [1]. According to the latest statistic, oral diseases accounted for approximately 3.5 billion cases in 2019 alone, with caries in permanent teeth being the most common dental problem. It is also estimated that 520 million children worldwide suffer from primary tooth decay, while 2 billion adults have permanent tooth decay [2]. While caries affects all age groups, periodontitis (gum and alveolar bone disease) exhibits increasing prevalence for adults between 30–40 years old [3]. On the other hand, dry mouth, also known as xerostomia, is a common problem for older adults affecting 22% of people worldwide [4]. Xerostomia is defined as the subjective experience of dry mouth complained by the patients, whereas hyposalivation refers to an objectively measured low salivary flow. These two distinct and independent phenomena may manifest separately or concurrently. The topical application of artificial saliva to replace or trigger the production of natural saliva is one of the promising treatments for xerostomia [5]. Therefore, accurate diagnosis and monitoring would allow dentists to recommend proper initial treatments to prevent the recurrence of dental problems and reduce unnecessary overhead costs from over-treatments. The current state-of-art for dental diagnosis and monitoring is heavily dependent on physical examination, patient's medical history and dental imaging (e.g. orthopantomogram and single-tooth radiography) [6]. However, current techniques are limited to discovering past and present dental problems. Therefore, they do not contribute to early detection for preventive treatments.

Microfluidic compact disc technology for rapid analysis of human physiological fluids, especially blood, is a growing technology in the clinical arena [7]. However, similar technology is less explored for other physiological fluids such as saliva, sweat and urine. Microfluidic technology could offer a simple and robust design for rapid and high-throughput salivary detection. The realization of this technology can be useful to facilitate the preventive diagnostic of oral diseases in a rapid and inexpensive manner [8]. Recently, Baumgartner and co-workers developed an automated microfluidic compact disc (OralDisk) as a non-invasive molecular-based platform for whole saliva analysis and detection of ten oral disease-causing bacteria (i.e. periodontitis- and caries-associated bacteria) [6]. Despite the anticipated advantages of the technology, further advancement for future non-invasive monitoring of oral diseases is challenging due to several shortcomings. First, saliva is a complex bodily fluid that contains various components (e.g. electrolytes, proteins, enzymes, nitrogenous products), high viscosity due to a high abundance of glycoproteins, and surface tension across different individuals which complicate the microfluidic-based sample analysis. Saliva samples may also contain bulky food debris, traces of cells and glycoproteins that could clog the microfluidic channels [9]. Johannsen, Müller [10] introduced an automated pretreated system of the whole saliva using the magnetic-beating method. The introduction of magnetic force to mix salivary samples in the preprocessing stage successfully reduced saliva viscosity from 10.4 to 2.3 centipoise (cP) within 4 min for subsequent point-of-care (POC) protein analysis. According to Shi, Ye [11], the current standard requires several hours of the processing time prior to sample analysis. The ability of reported magnetically influenced centrifugal microfluidic to pretreat samples under 5 min is a huge leap to centrifugal microfluidic advancement. Park and colleagues proposed a novel disc design by interconnecting three reaction chambers preloaded with biomarkers functionalized beads for common processes (i.e. sample loading, incubation and washing). The new disc design reduced the number of valves yet effectively accommodated multiplex immunoassays for various physiological fluids such as whole saliva and blood. In addition, centrifugation at

3600 rpm for 2 min successfully eliminated salivary mucus and reduced its viscosity for accurate salivary analysis and detection [12].

To date, various sensing methods such as optical (e.g. chemiluminescence, Raman spectroscopy, IR spectroscopy), mechanical and electrochemical (e.g. electrochemical impedance spectroscopy) are arguably most compatible with the miniaturized microfluidic compact disk (CD) platform [9, 13–16]. Many microfluidic-based platforms adapted the electrochemical impedance spectroscopy (EIS) for better opportunities to develop highly sensitive biosensing platforms. EIS is a method with a wide range of uses, including the detection of biomarkers and cellular deformities in physiological fluids, detection of food-borne pathogens and cancer cells, as well as the investigation of metal corrosion and coated metal surfaces [17]. In the dentistry field, EIS spectroscopy is primarily used for the examination of dental materials, corrosion and mycosis identification [18–21]. EIS analysis of saliva is limitedly studied despite promising potential as a diagnostic tool for oral and systemic diseases. Recently, Lu and colleagues adapted the EIS method to evaluate dehydration levels and kidney function. They reported that the measured conductance value of saliva was consistent with the serum osmolality, which was also associated with dehydration levels. The promising finding could serve as an indicator for simultaneous monitoring of dehydration level and kidney function [22]. From this perspective, the use of integrated microfluidic CD and non-invasive salivary fluids appears to be a promising approach for advanced monitoring of oral diseases.

Saliva is a candidate for analysis that is becoming more and more popular because it is simple to collect in tubes, and its collection is non-invasive. In comparison to many other medical specialties, such as infectious diseases like respiratory tract, bloodstream, and gastrointestinal infections, where point-of-care or near-patient systems are already commercially available or in the product development stage, the development and application of chairside molecular diagnostics in the field of oral health lags. Although the cost of treating dental diseases in EU member states reached €90 billion in 2015, and there were additional productivity losses of over €50 billion, oral diseases still carry a heavy socioeconomic burden [23]. Furthermore, the established link between periodontal disease and other systemic illnesses has begun to increase public awareness of the significance of oral health, particularly early detection, prevention, and post-treatment monitoring [24]. Saliva has also been used in non-oral diagnostic procedures to look for biomarkers or 481 nucleic acids associated with illnesses like Type 2 diabetes mellitus, cardiovascular conditions, and Alzheimer's disease [25].

Studies that have used the electrical characterization of saliva itself are very rare in the literature. Recently, a biodevice for measuring salivary conductivity with miniaturized sensing probes has been created and applied in saliva conductivity analysis in healthy adults and patients with chronic kidney disease [22]. A disposable printed-circuit-board (PCB) electrode and the use of highly biocompatible, stable, and reusable gold as the conductive material were the main features of this portable sensing system. A saliva sample with an incredibly small volume (50 μL) could be used for the conductivity test thanks to the co-planar design of coating-free gold electrodes. This completely goes in line with the attempts to make collecting saliva easier, eliminate the need for trained personnel and improve compliance. In this series of research, it has been demonstrated that in dehydrated healthy adults, an increase in salivary conductivity was linked to an increase in serum and urinary osmolality. Furthermore, it has also been found that salivary conductivity and age had strong correlations [26]. In this case, reduced saliva secretion due to xerostomia or dry mouth syndrome will influence the saliva's compositions and conductivity. Therefore, it is interesting to investigate the electrochemical property of commonly used medication (i.e. artificial saliva) in the treatment of xerostomia along with its influence on commonly used mouthwashes. Centrifugal microfluidic has been successfully adapted in the industry with a recent commercial innovation by Abaxis Inc

through their Piccolo Xpress for routine blood chemistry diagnostic integrated with a specially designed analyzer. In this innovation, the high-throughput microfluidic disc could generate results within 12 minutes [27]. Similarly, our proposed fabricated disc could be translated into salivary-based microfluidic system for analysis of oral diseases. The realization of the microfluidic-based products to reach end users depends on several key parameters such as robustness of assay, accurate detection of target biomarkers, careful deliberation of the design system, hardware development, disc assembly as well as minimal production and assay costs [28].

This paper presents a novel systematic design of a microfluidic CD to investigate electrochemical property changes between artificial and real salivary samples mixed with different types of commercially available mouthwashes (*i.e.* chlorhexidine-, fluoride- and essential oil-based mouthwashes) using electrical impedance analysis. The microfluidic CD design is composed of multiple sets of targeted saliva and mouthwash chambers connected to a mixing chamber for subsequent multiplex electrical impedance analysis. We also introduce a new low-cost and rapid fabrication of microfluidic CD using xurographic technique. The present concept design and fabrication of the CD platform offer several advantages such as (i) fully automated fluid handling (i.e. loading, incubation, mixing and analysis), (ii) low sample volume (10–20 μL, (iii) dynamic analysis with better analysis. This study introduces new avenues of point-of-care theranostic centrifugal microfluidic CD platform for prevention of oral diseases and determination of adjunctive treatments using specific mouthwashes.

## Materials and method

### Microfluidic compact disc (CD) fabrication using xurographic technique

In this research, the novel design of a microfluidic CD was developed using polyvinyl chloride (PVC) foils using the xurographic technique. A total of seven PVC foil layers made up the microfluidic CD. The CAD designs of layers are shown in Fig 1. Layers 1, 6 and 7 were made using 80 μm PVC foils, while layers 2,3,4 & 5 were made with 250 μm PVC foils. Layer 1 serves as the top cover with alignment and inlet holes. Layer 6 holds the aluminium electrodes and layer 7 is the bottom cover with an aluminium connection to connect each electrode in layer 6 to the outer end of the disc. Layer 2 has full microfluidic channels, chambers and venting channels. Layer 3 & 4 has microfluidic chambers and channels. Layer 5 is only made with chambers. Each PVC foil has an adhesive coating that enables the foils to bond to each other when laminated with heat.

The designs were made using Autodesk AutoCAD 2021 software and transferred into Graphtec Cutting Master software to sequence the plotting in the cutter plotter. Graphtec CE6000-60 plus cutter plotter (Graphtech America, Inc. Irvine, CA, USA) with a 45˚ cutting blade (CB09U) and a cutting mat (12" Silhouette Cameo Cutting Mat, Sacramento, USA) were used to cut the PVC foils according to the sequence made in Cutting Master. The process is similar for layers with aluminum electrodes and connection routing (layers 6 & 7), except that each PVC foil was applied with an aluminum tape before the cutting was sequenced. Upon completion of the cutting sequence, the PVC foils were removed, and the excess aluminum parts were carefully peeled off the PVC foil. Each layer was then carefully stacked and aligned according to Fig 1. To ensure a good alignment, a few cylindrical pins with 1 mm diameter were inserted through the alignment holes in each layer. Scotch tapes were used to hold the layers in place temporarily during the lamination process. The stacked layers were inserted into the thermal laminator at 200˚C for approximately 10 seconds.

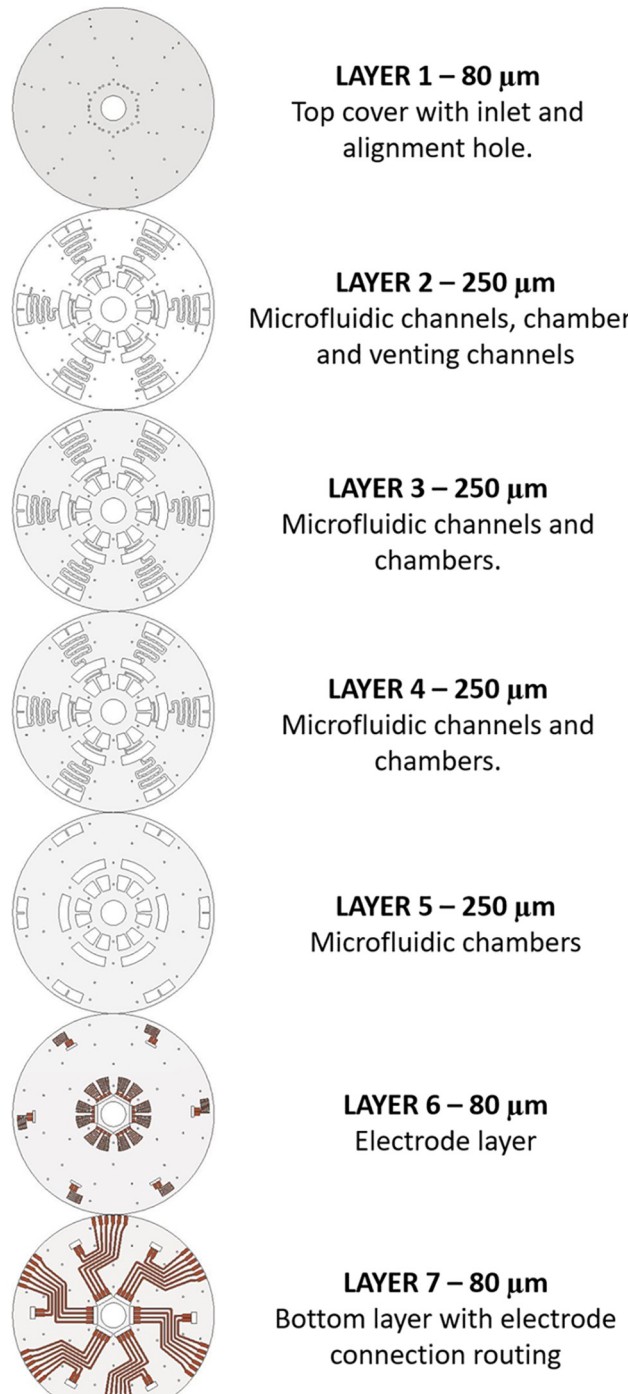

**Fig 1. Schematic illustration of microfluidic CD layers.**

## Theory of operation

The microfluidic CD was designed with six identical microfluidic sets to run six simultaneous or repeated tests. Every set consists of four parts. The design was optimized for minimal sample volume, lower rotational speed, and CD space. The final design is shown in Fig 2. In this

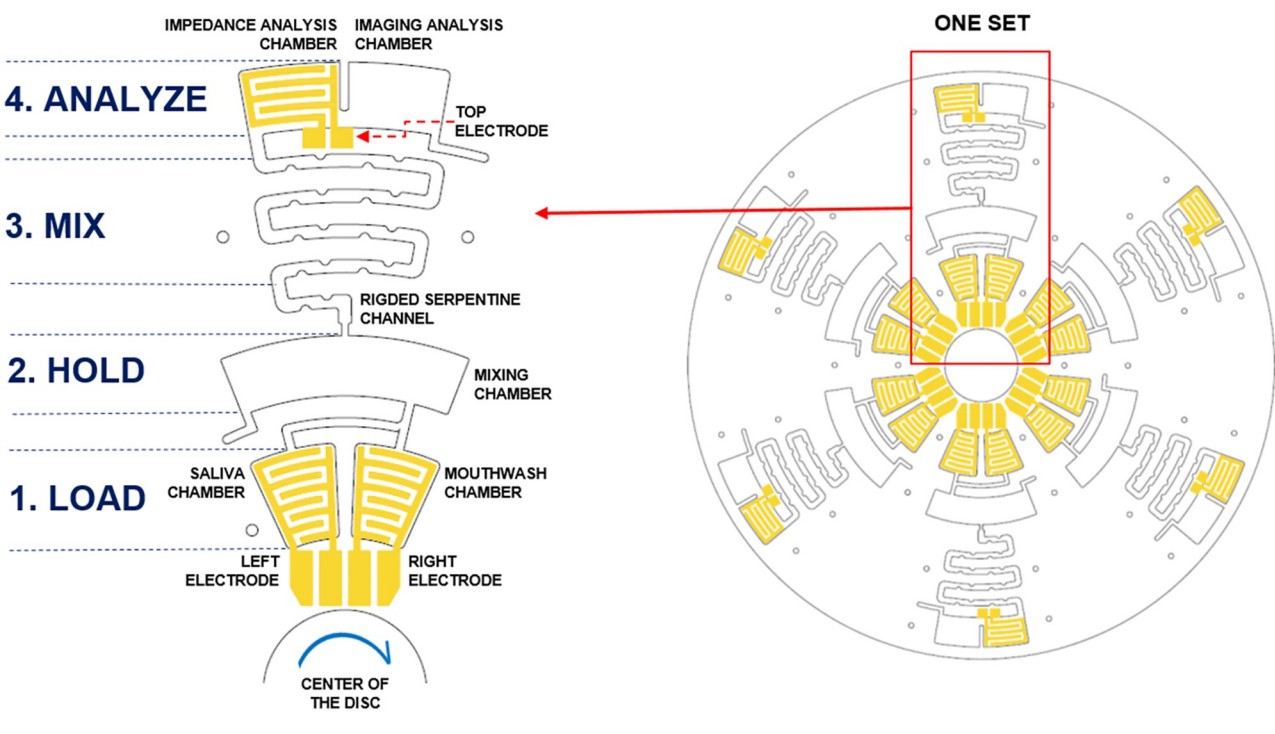

**Fig 2. Working design of the microfluidic CD.**

microfluidic CD design, three sets were assigned for mixing the sequence of real saliva with three different types of mouthwashes. Another three sets were assigned for artificial saliva mixed with three different types of mouthwashes. Real saliva used in the subsequent analysis was taken from the same individual. The sequencing and functionality of each part were divided into four stages; (1) LOAD, (2) HOLD, (3) MIX and (4) ANALYZE, as illustrated in Fig 2. In the first stage (LOAD), the mouthwashes MW1, MW2 and MW3 were mixed with either artificial saliva (AS) or real saliva (RS). Then, the disc was spun to a speed until both solutions entered the HOLD chamber. The speed was slowly increased until the solution in the HOLD chamber burst into the serpentine channel (MIX). The serpentine channel was designed with ridges to enhance the mixing efficiency. The ANALYZE chamber (final stage) was designed to equally separate the solution into two volumes covering the right and left sides of the chamber. The right side with a transparent bottom layer was for sample imaging, whereas the right side was fixed with an embedded electrode for impedance measurement. The formulations containing artificial saliva or real saliva mixed with different types of mouthwashes tested on specialized chambers were listed as follows:

a. Chamber T1 –RS + MW1

b. Chamber T2 –RS + MW2

c. Chamber T3 –RS + MW3

d. Chamber T4 –AS + MW1

e. Chamber T5 –AS + MW2

f. Chamber T6 –AS + MW3

## Preparation of tested liquids

Artificial saliva (AS) composed of carboxymethyl cellulose was prepared according to the recipe of the Pharmacy Institution Belgrade (registered under the Republic of Serbia's master preparations) and was subjected to several analyses in this study. Real saliva (RS) was collected from one healthy individual. The following listed three different types of mouthwashes investigated in this study:

1. 0.12% chlorhexidine-digluconate solution, PerioPlus Protect (PP) (Curaden International AG, Switzerland), abbreviated as MW1;

2. Elmex, mouthwash (FL) (Gaba International, France) containing active ingredients of 100 ppm amine fluoride and 150 ppm sodium fluoride, abbreviated as MW2;

3. Listerine (LT) (Johnson & Johnson, New Brunswick, NJ, Cool Mint Listerine), abbreviated as MW3.

## Saliva collection

Three healthy volunteers (two females and one male) were recruited in this study. The participants were volunteer staffs from the Department of Dentistry, University of Novi Sad. This study has obtained approval from the Ethical Committee of Dentistry Clinic of Vojvodijna Clinic, Faculty of Medicine, University of Novi Sad. In addition, the recruited volunteers gave their informed consent. The selection criteria and salivary collection protocol were based on the recommendations from Bhattarai, Kim [29]., The inclusion criteria were both genders, participants younger than 65 years, self-declaration of no general diseases and no pregnancy and especially no symptoms affecting the salivary glands or the oral mucosa.. Pregnant women, persons who complain of dry mouth or eyes, participants with oral ulcerations or other contact sensitivities, people with autoimmune illnesses, people on short-term or long-term drugs known to promote oral dryness were all excluded from the study.

The saliva was collected using the spitting method, and the testing were completed within 2 hours after the collection. The spitting method comprised collecting saliva on the tongue and spitting into a graded test tube every 30 to 60 seconds. The whole sampling procedure comprised obtaining the specimens by spitting saliva into a test tube, transferring the material to a syringe, and subsequently injecting the syringe into the chip. About 5 ml of unstimulated saliva was collected and stored at a temperature of 4˚C in plastic before the analysis.

## Method of measurement

Each set consists of three aluminum electrodes labelled in Fig 2; Right, Left and Top. Electrical continuity was first conducted to ensure that the electrode path was fully connected as per designed with no short-circuit in between the electrode legs. The impedance analysis was conducted upon completion of electrical continuity tests. The impedance measurement was conducted using HIOKI IN3590 Impedance Analyzer with a frequency sweep ranging from 1 to 200 kHz (250 data points). Upon completion of reference (open-circuit) measurement, 20 μL of mouthwash was added into the mouthwash chamber and 20 μL of saliva sample was added into the designated saliva chamber. The impedance, capacitance and conductance measurements were taken on each left and right electrode. Once the solution entered the ANALYZE chamber, the disc was stopped, and impedance, capacitance and conductance measurements were taken from the top electrode. An image was also taken with a focus on the Image Analysis Chamber for subsequent analysis.

For the statistical analysis of the effect of the 3 different mouthwashes on both artificial and real saliva, the non-parametric one-way Anova (Friedman) with post hoc test was used. The level of significance was set at $p < 0.05$.

## Results and discussion

The outcome of xurographic fabrication in this study is the microfluidic CD. Aluminium electrode traces are 1 mm wide and interdigitated electrodes have a trace width of 0.55 mm and an interelectrode gap of 0.55 mm. The layers were laminated to complete the microfluidic CD (Fig 3). Microfluidic CD provides rapid, automated, and multiplex analysis of saliva samples. In this application, we demonstrated simultaneous microfluidic mixing and electrical impedance analysis of real and artificial saliva with three different mouthwashes (total of six analysis) in a CD executed in a single run. The built-in aluminum electrodes enable impedance characterization of liquids before mixing and after mixing.

The results of the investigation of the electrochemical impedance changes between artificial and real salivary mixed with three different types of mouthwash are shown in Figs 4–7. Fig 4(a) shows the conductance of real saliva from three participants with each subject measured for 9 times (3 set of electrodes on a CD run for three times) from 1 kHz to 200 kHz showing consistent results across different electrodes and number of runs. The results showed that artificial saliva has a higher conductance (G = 15 mS, SD = 0.12, n = 9) compared to real saliva (G = 6 mS, SD = 1.35, n = 27) at 200 kHz. This data can be explained by the fact that saliva samples are real clinical samples that contain, in addition to the basic ingredients of saliva, proteins, enzymes and electrolytes, and over 700 species of bacteria, their metabolites, gingival sulcus content and food debris. Such a complex composition certainly contributed to the reduced conductance values. Fig 4(b) shows three types of antiseptic (mouthwashes) solutions for the frequency range from 1 kHz to 200 kHz. The analysis of the conductance of antiseptic solutions speaks in favor of the highest value of the conductivity of the fluoride solution (3 mS, SD = 0.35) in relation to the solutions of Listerine (0.75 mS, SD = 0.01) and a chlorhexidine (0.41 mS, SD = 0.01) at the maximum frequency of 200 kHz. The presence of free and active fluoride ions certainly significantly contributed to higher values of fluoride solution conductance compared to the remaining two analyzed types of mouthwash.

Fig 4(c) illustrates the mixtures of real saliva from three healthy participants and artificial saliva with the three different types of mouthwash in the frequency ranges from 1 to 200 kHz. The mixture of saliva (both real and artificial) with mouthwash resulted in lower conductance than pristine saliva as conductance values of mouthwashes is lower than that of saliva. Specifically, the conductance values for the mixture of artificial saliva with Listerine experienced the lowest conductance G of 6.54 mS (SD = 0.15) at 200 kHz. This is followed by conductance values of artificial saliva mixed with chlorhexidine at (G = 7.54 mS, SD = 0.13) and artificial saliva mixed with fluoride mouthwash at (G = 9.43 mS, SD = 0.28) at 200 kHz. For real saliva, mixing with mouthwashes also reduces the conductance and shows consistent trends across three separate subjects with fluoride solution giving highest conductance (G = 4.80 mS, SD = 0.37), followed by Listerine (G = 3.46, SD = 0.52) and Chlorhexidine (G = 2.69 mS, SD = 0.64) from 27 measurements for each mouthwash. Higher conductance values for mixtures of fluoride solutions can be attributed to the amount of active fluoride ion. Studies examining the electrical parameters of complex physiological fluids are rare in the literature. Comparing the measured capacitances for water samples reveals that the capacitances drop as the frequency rises. When the frequency is increased from 100 Hz to 2 kHz, the measured capacitance for saltwater decreases from 7.98 F to 4.11 F [30].

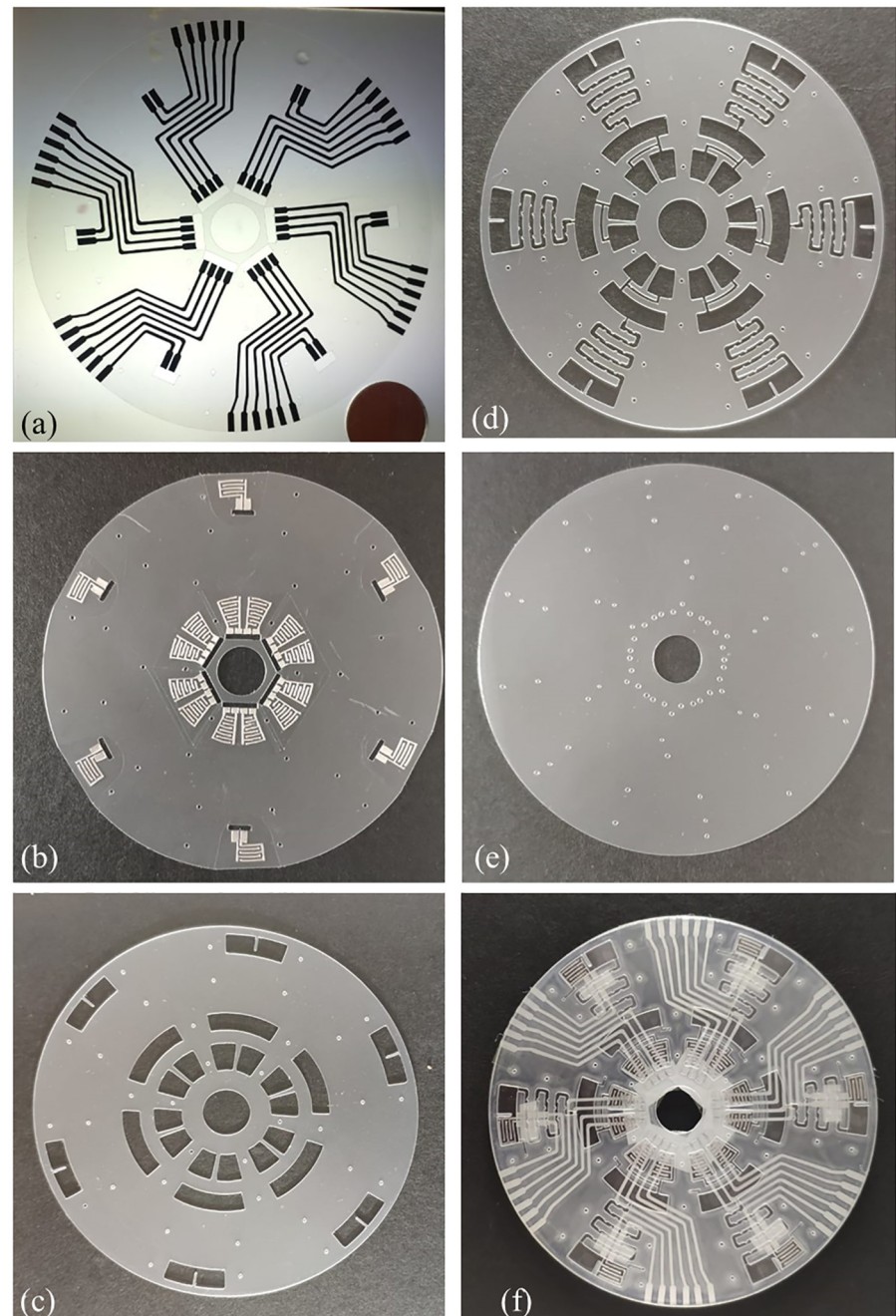

**Fig 3.** Fabricated microfluidic CD (a) bottom layer with conductive trace, (b) aluminum interdigitated electrode layers, (c) microfluidic chamber layer, (d) layer with microfluidic chambers and channels, (e) top cover layer with fluidic inlet and (f) assembled and bounded microfluidic CD.

Fig 5a–5c show that the capacitance value decreases with increasing frequency in all tested liquids and liquids with higher conductivity also show higher capacitance values. Our tested fluids are mainly electrolytes. Fluids' ionic polarization cannot keep up with changing electric field polarization as frequency increases. Hence, capacitance falls as the frequency of the electric field increases. Studies investigated the capacitance for salt water, regular water, and

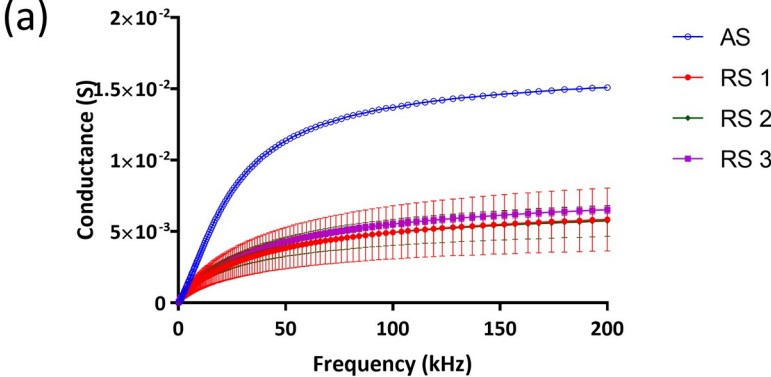

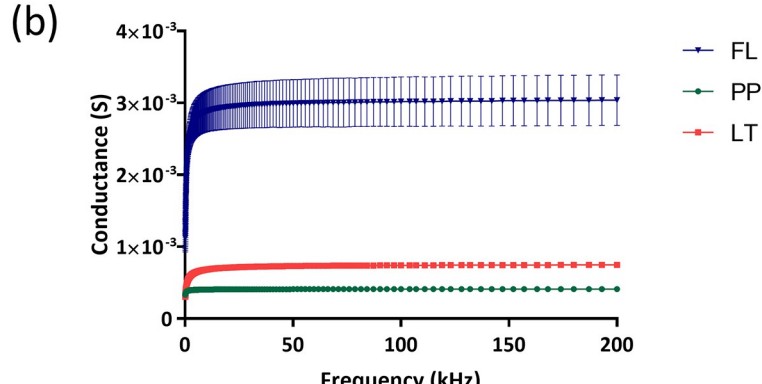

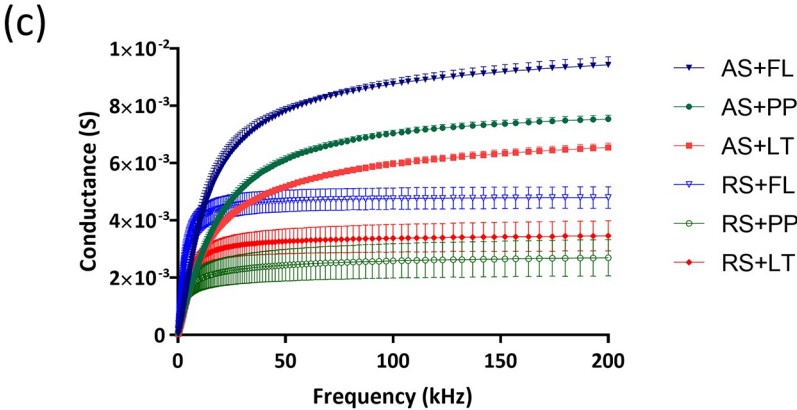

**Fig 4.** Conductance values of (a) real saliva from three healthy subjects (RS1, RS2, RS3) and artificial saliva (AS), (b) chlorhexidine (PP)-, fluoride (FL)- and essential oil (LT)-based mouthwashes, and (c) mixed liquids with real saliva and artificial saliva.

mineral water uniformly reported a decrease as the frequency rises [31]. Similar to our findings, they also reported that more capacitance is present in liquids with high electrical conductivity. By analyzing all the obtained results, it is evident that of all the examined individual liquids, but also the mixture of saliva solutions with mouthwashes, the capacitance of the fluoride solution is the largest and significantly deviates from all other analyzed liquids. These can

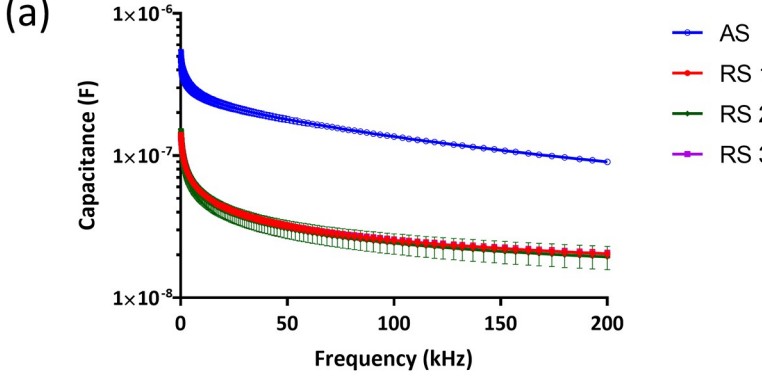

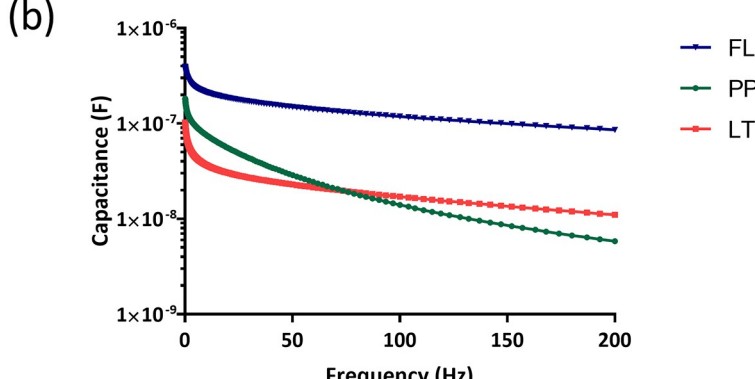

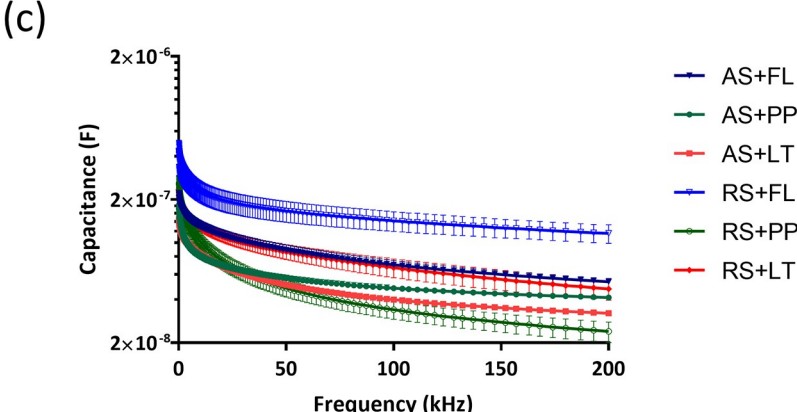

**Fig 5.** The capacitance values of (a) real saliva (RS) and artificial saliva (AS), (b) chlorhexidine (PP)-, fluoride (FL)- and essential oil (LT)-based mouthwashes, and (c) mixed liquids.

be attributed again to the presence of a large amount of free and reactive fluoride ions in said solution. Also, this may speak in addition to caution when recommending the use of fluoride in patients who have restorations made of different metals in their oral cavity. In accordance with the previous finding, the lowest values of capacitance were recorded in the mouthwash solution containing chlorhexidine, as well as its mixtures with real and artificial saliva, as shown in Fig 5(c) [30, 31].

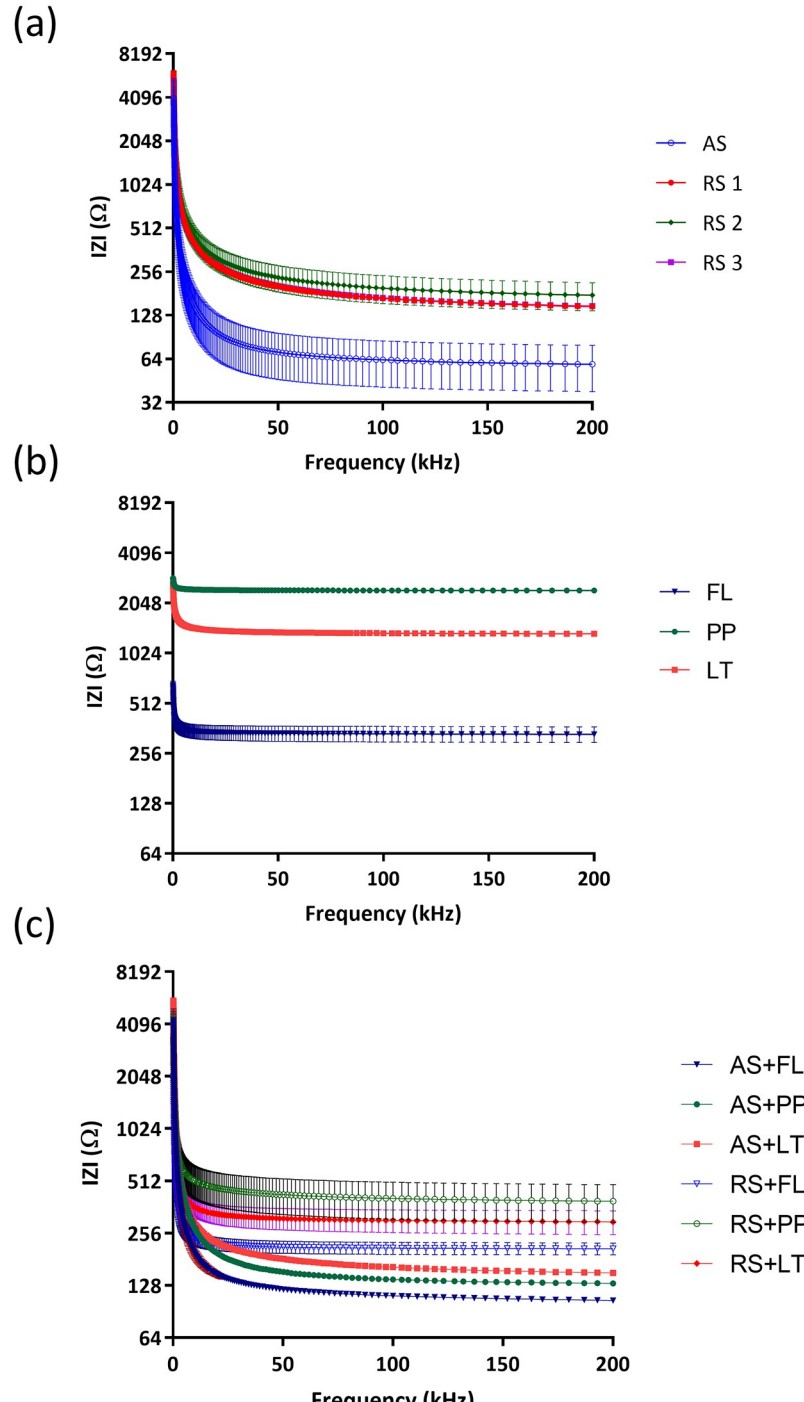

**Fig 6.** |Z| value of (a) real saliva (RS) and artificial saliva (AS), (b) chlorhexidine (PP)-, fluoride (FL)- and essential oil (LT)-based mouthwashes, and (c) mixed liquids.

Fig 6a–6c show the impedance response of saliva (artificial and real), mouthwashes and mixed liquids containing saliva (real or artificial) and different types of mouthwashes with respect to frequency. Impedance of real saliva is consistently higher than artificial saliva (Fig 6 (a)), although taken from the three different individuals. Fig 6(b) represents the impedance

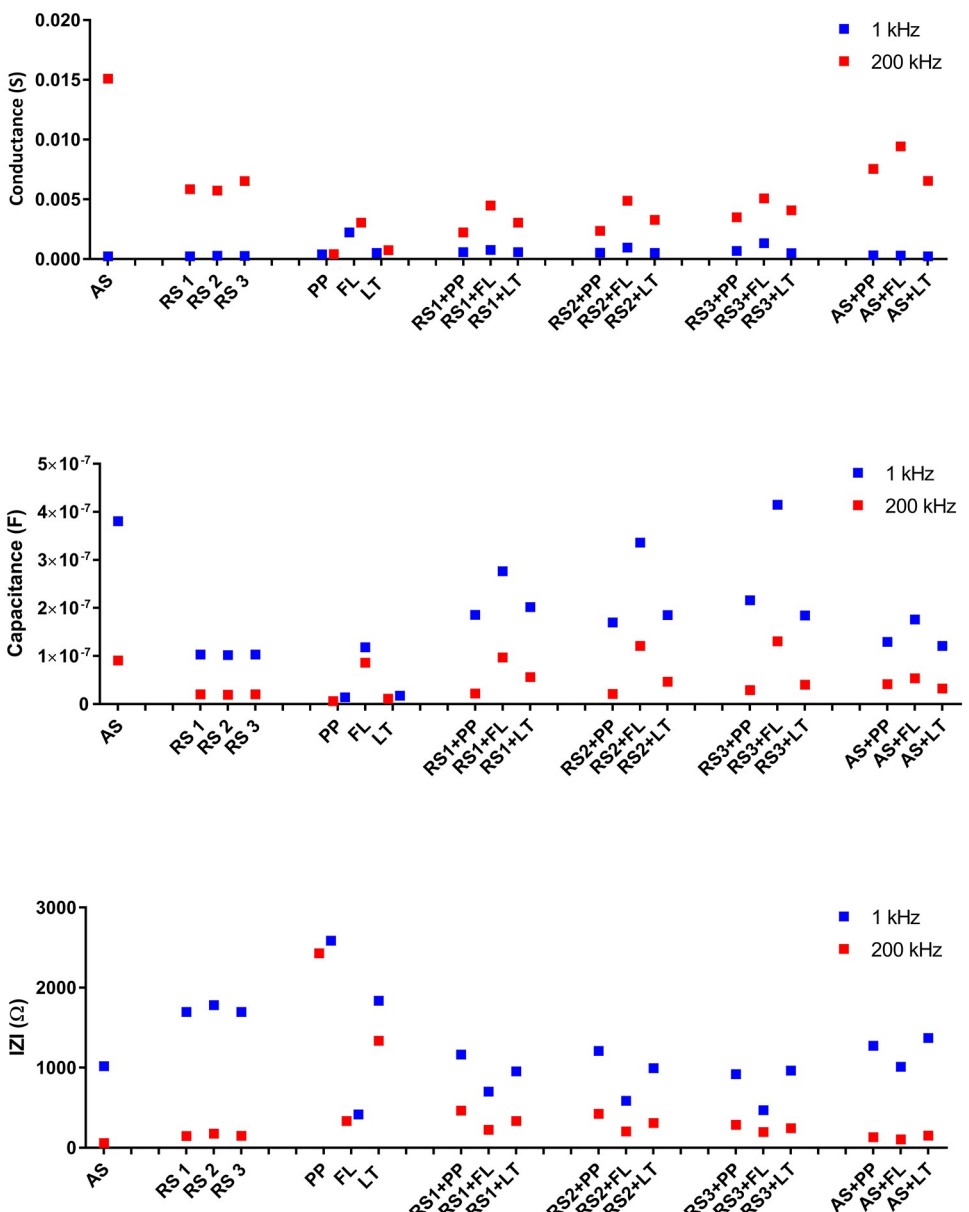

**Fig 7.** (a) Conductance, (b) capacitance and (c) |Z| values of real saliva from 3 subjects (RS1, RS2, RS3) and artificial saliva (AS), mouthwashes (PP, FL, LT) and the mixed liquids tested at 1 and 200 kHz.

response of three different types of mouthwash, the impedance for all the mouthwashes decreases as the frequency increases. Chlorhexidine, Listerine and fluoride mouthwashes showed the highest, moderate and lowest impedance response with increasing of frequency, respectively. The lower impedance of fluoride could be due to the presence of sodium fluoride [32]. As shown in Fig 6(c), real saliva mixed with fluoride-based mouthwash had the lowest impedance at Z = 208 Ω (SD = 16.35, n = 27) respectively at 200 kHz, whilst real saliva mixed with Listerine has impedance of 297 Ω (SD = 46, n = 27) and real saliva mixed with chlorhexidine has highest impedance of 391 Ω (SD = 95, n = 27) measured at 200 kHz. The increase pattern of impedance has been observed in all mixed liquids across the broad in agreement with

higher impedance values of mouthwashes shown in Fig 6(b). In Fig 6(c), it is observed that chlorhexidine-based mouthwash has a slightly higher impedance which also correlated to saliva behavior shown in Fig 6(a).

Fig 7(a) shows the values of conductance at frequencies of 1 kHz (blue) and values of frequencies of 200 kHz (red). Since the capacitance has maximum at a lower frequency, 1 kHz and conductance (G) has maximum at a higher frequency; hence, these frequencies are chosen for further analysis. Microfluidics based automated mixing enables efficient and consistent mixing of saliva with mouthwash as demonstrated by conductance values from three separate subjects. At 200 kHz, mixing real saliva with Listerine, chlorohexidine-based and fluoride-based mouthwashes resulted in conductivity reduction of 42%, 55% and 20% respectively. This decrease was statistically significant for all 3 used mouthwashes ($p < 0.0001$, one-way Anova test). We have also found statistically significant differences in conductance, impedance, and capacitance values after mixing of mouthwashes with real and artificial saliva ($p < 0.0001$, one-way Anova test). The conductance of all analyzed fluids at a frequency of 1kHz is in the range below 1 mS, while at frequencies of 200 kHz, these values are in the range from 2 mS to 15 mS (see Fig 7a). The conductance value increases as ion concentration increases due to a greater degree of ionic dissociations [33]. In addition, lower charge accumulations occurring around the electrodes at a higher frequency may lead to an increasing number of mobile ions. As a result, conductivity increases with frequency. The obtained values of conductivity at higher frequencies fully correspond to the values of salivary conductivity given in the literature, where lower values of conductivity can be considered values of 4.84 ± 1.01 μS/cm and higher values of 7.994 ± 1.02 μS/cm [26]. The value of conductance can be obtained by applying the following equations:

$$G = \frac{1}{R} \tag{1}$$

G refers to the experimental conductance value, and R is defined as resistance.

$$k = G \times cell\ constant \tag{2}$$

where k is defined as the specific conductivity whereas cell constant is a quantity describing the geometry of the cells:

$$cell\ constant = \frac{l}{A} \tag{3}$$

where *l* is the distance between the electrodes and A refers to the common electrode area.

Fig 7(b) shows a higher capacitance value for all tested samples (i.e. real saliva, artificial saliva, mouthwashes and mixed liquids) at a low frequency (1 kHz). At lower frequencies, a phase difference arises between the electrodes due to two dielectric phenomena that occur concurrently; dipoles align in the direction of the applied field, while ions flow towards the electrodes (Maxwell-Wagner-Sillars, MWS) phenomenon. As the frequency increases, higher charge accumulations occur at the electrode and material interface leading to greater ion mobility and an increase in conductivity. As a result, capacitance decreases [34]. Fig 7(c) shows the impedance response of our tested liquids at 1 kHz and 200 kHz. The impedance is higher for mixed liquids as compared to the saliva at a lower frequency. This confirms the presence of higher ions in the mixed liquids due to higher ionic concentrations contributed by the mouthwashes.

During the experiment, Listerine showed a significant greenish color before mixing. Upon mixing with real saliva, the mixed liquid turns pale green. To understand the homogeneity of mixing product, histogram analysis was conducted to visualize the red, green, and blue (RGB)

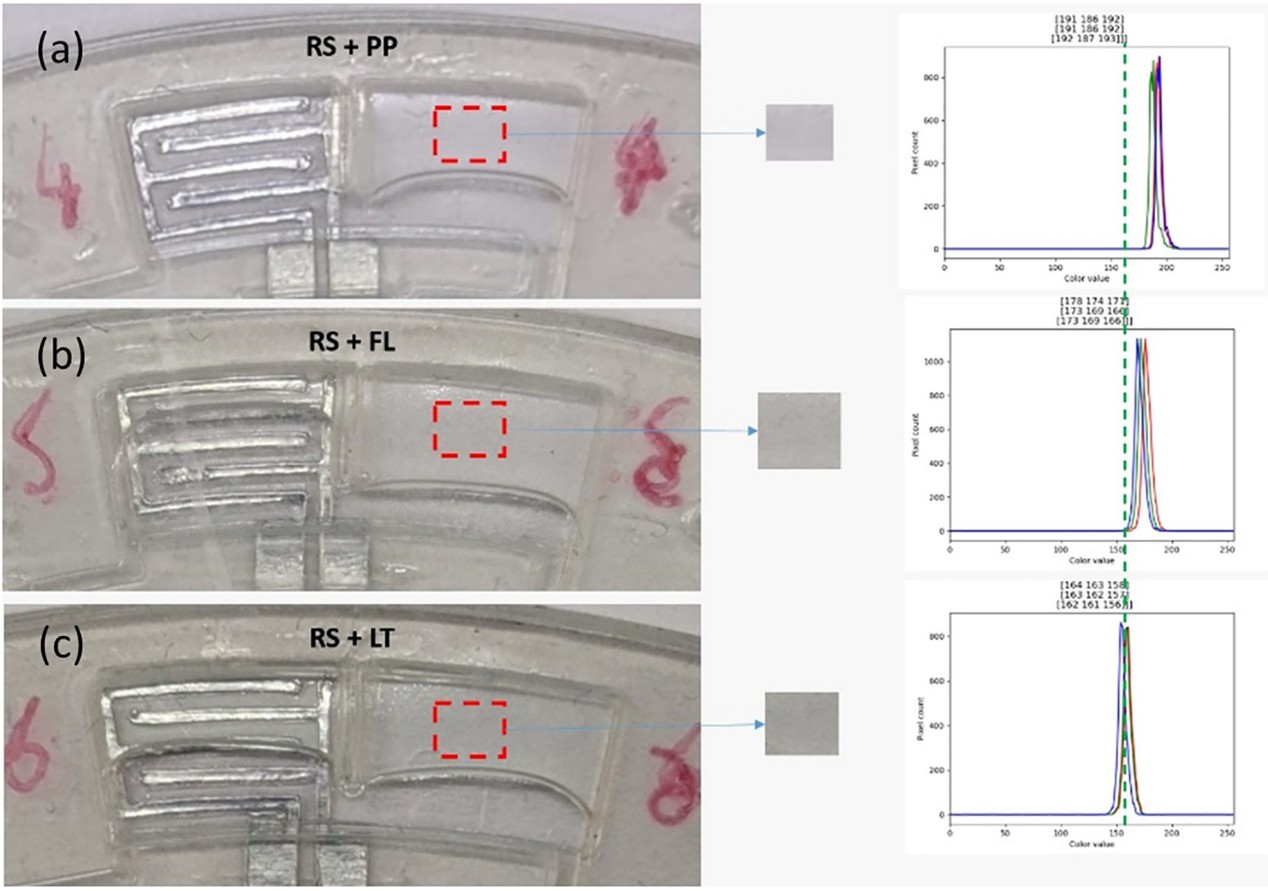

**Fig 8.** Mixing of real saliva (subject 1) and different mouthwashes (a) chlorhexidine-, fluoride-based mouthwashes and (c) Listerine and their respective RGB histogram analyses.

components in each captured image. In Fig 8, the peaks of RGB are observed to be closer with very steady lines and no RGB components are observed to be existing separately which indicates a coherent mixing. Even though it cannot be clearly observed with naked eye, histogram results show the difference in the RGB peaks between mixed liquid containing real saliva and chlorhexidine (Fig 8a) and mixed liquid containing real saliva and Listerine shown in Fig 8(c). Fig 8(a) is more transparent and whiter, whereby Fig 8(c) is greener from RGB analysis.

Results from Fig 6 are important in giving the information to the clinician. Clinically, this could implicate that patient with xerostomia usually have higher conduction. Precaution should be taken when prescribing fluoride solution to this patient because they have a low pH and high conductivity [35]. The use of oral care products such as mouthwashes and toothpaste are general recommendations to maintain good oral hygiene. Mouthwashes are commonly divided into three different classifications, namely antiseptic, plaque-inhibiting and preventive. Chlorhexidine (CHX)-based mouthwashes are considered the gold standard of antiseptic mouthwashes due to their effectiveness in combating bacteria, spores, and fungi. In this sense, antiseptic-based mouthwashes can be helpful in reducing nosocomial infections at different care levels (i.e. preoperative, intraoperative and post-operative) of oral-surgical procedures. Fluoride-based mouthwashes may be suggested or prescribed for patients over the age of eight with a higher risk of developing dental caries due to hyposalivation problems or patients

undergoing orthodontic treatment as it can be challenging to maintain good oral hygiene while wearing a fixed appliance [36]. Fluoride-based types of mouthwash in various forms such as sodium fluoride (NaF), stannous fluoride (SnF$_2$) and amine fluoride (AmF), have shown promising caries control and as an efficient addition for mechanical plaque removal [37, 38]. A commonly available essential oil containing mouthwash called Listerine has a fixed formula that includes 0.064% thymol, 0.092% eucalyptol, 0.042% menthol, and 0.06% methyl salicylate. Recent studies confirmed that Listerine reduces gingival inflammation and dental plaque by disrupting membranes at high concentrations and inactivating vital enzymes at lower concentrations, respectively [39, 40]. Evidence from at least 26 reported studies evaluated by Alshehri [41] showed clinical efficacy of Listerine for short-term (<3 months) and long-term (3–6 months) usage due to anti-plaque and anti-gingivitis properties. This further support the use of essential oil-based mouthwashes (Listerine) in the daily oral regimen to maintain personal oral hygiene.

This CD microfluidic platform can be integrated into various clinical applications for rapid POC test. First, by applying this system, it would be possible to conduct clinical studies with minimal amounts of salivary analytes in a faster and more efficient way. Then, at the level of individual point of care applications, with this study we have shown that the analysis of the electrical properties of saliva is feasible and simple, and each of the analyzed parameters is clinically relevant, both for healthy patients and for patients with oral diseases and disorders in the composition of saliva.

In the conducted study, the saliva of healthy volunteers, without oral and general diseases was used, so the obtained values were in the physiological range, as expected. Future clinical studies will, we believe, confirm the effectiveness of this system in evaluating the electrical properties of saliva in patients with various disorders. However, even in healthy patients at low risk for developing oral diseases, it is possible to assess which type of the most commonly used mouthwash solutions lead to the most favorable changes in the composition of saliva, with the least chance of adverse effects.

Finally, in patients with reduced salivary secretion, it is possible to evaluate the effect of the artificial saliva that is used, as well as the combination of artificial saliva and antiseptic solutions that will reduce the symptoms, complaints and reduce the risk of oral diseases occurrence in patients with xerostomia. The proposed CD microfluidics platform can be expanded further for clinical applications using optical and electrical impedance analysis as demonstrated in this research.

## Conclusions

This paper has presented a fully integrated microfluidic CD consisting of six unique chambers for multiplex immunoassay from sample loading, holding, and mixing to the final detection of samples. The adaptation of the xurographic technique demonstrated a fast, low-cost and simple fabrication strategy of our designed microfluidic CD. Our microfluidic CD was able to detect changes in electrical impedance of physiological fluids and different types of mouthwashes. This conceptual study could serve as rapid detection of oral diseases with accurate information for effective early preventive treatments. The performance of the fabricated microfluidic CD was investigated on real and artificial saliva, as well as three types of mouthwashes containing different active components (i.e. chlorhexidine digluconate, fluoride and essential oils). In this study, artificial saliva showed higher conductivity (15 mS) compared to real saliva (6 mS) at 200 kHz. Furthermore, fluoride-based mouthwash has the highest conductivity (3 mS) among the tested mouthwashes, followed by Listerine (0.75 mS) and a chlorhexidine (0.41 mS) when measured using integrated electrodes from microfluidic CD. Mixing of

saliva and mouthwashes reduces electrical conductance up to 50% and changes in electrical properties are found to be statistically significant. This suggests the ability of the fabricated microfluidic CD to differentiate different types of samples, thus proving its detection accuracy. Based on the electrical impedance changes of the tested liquids (i.e. real saliva, artificial saliva or mixed liquids), it was found that the mixed liquids containing saliva and fluoride-based mouthwash showed an observed change compared to other liquids. The findings of this study indicate that bioimpedance analysis can be used as an ideal tool for monitoring of saliva and identification of suitable preventive treatments (e.g. drug delivery, suitable mouthwashes) for various oral diseases. In addition to that, the results were comparable which demonstrate the applicability of our microfluidic CD for future oral disease theranostics using the rapid salivary test.

## Supporting information

**S1 Data.**
(XLSX)

## Author Contributions

**Conceptualization:** Aung Thiha, Karunan Joseph, Bojan Petrović, Sanja Kojić.

**Data curation:** Aung Thiha, Karunan Joseph, Bojan Petrović, Sanja Kojić.

**Formal analysis:** Aung Thiha, Fatimah Ibrahim, Karunan Joseph, Bojan Petrović, Sanja Kojić, Nuraina Anisa Dahlan, Nurul Fauzani Jamaluddin, Saima Qureshi.

**Funding acquisition:** Fatimah Ibrahim, Goran M. Stojanović.

**Investigation:** Aung Thiha, Fatimah Ibrahim, Karunan Joseph, Bojan Petrović, Sanja Kojić, Nuraina Anisa Dahlan, Nurul Fauzani Jamaluddin, Saima Qureshi.

**Methodology:** Aung Thiha, Karunan Joseph, Bojan Petrović, Sanja Kojić.

**Supervision:** Fatimah Ibrahim, Goran M. Stojanović.

**Writing – original draft:** Aung Thiha, Fatimah Ibrahim, Karunan Joseph, Bojan Petrović, Sanja Kojić, Nuraina Anisa Dahlan, Nurul Fauzani Jamaluddin.

**Writing – review & editing:** Fatimah Ibrahim, Goran M. Stojanović.

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
