## [Decision Letter · Decision Letter 0]

24 Oct 2022

PONE-D-22-24397A Novel Microfluidic Compact Disc to Investigate Electrochemical Property Changes Between Artificial and Real Salivary Samples Mixed with Mouthwashes Using Electrical Impedance AnalysisPLOS ONE

Dear Dr. Ibrahim,

Thank you for submitting your manuscript to PLOS ONE. After careful consideration, we feel that it has merit but does not fully meet PLOS ONE’s publication criteria as it currently stands. Therefore, we invite you to submit a revised version of the manuscript that addresses the points raised during the review process.

ACADEMIC EDITOR: Due to a minor revision provided by one reviewer and a rejection by another, the academic editor has suggested a major revision.==============================

We look forward to receiving your revised manuscript.

Kind regards,

Ajaya Bhattarai

Academic Editor

PLOS ONE

Journal Requirements:

3. PLOS requires an ORCID iD for the corresponding author in Editorial Manager on papers submitted after December 6th, 2016. Please ensure that you have an ORCID iD and that it is validated in Editorial Manager. To do this, go to ‘Update my Information’ (in the upper left-hand corner of the main menu), and click on the Fetch/Validate link next to the ORCID field. This will take you to the ORCID site and allow you to create a new iD or authenticate a pre-existing iD in Editorial Manager. Please see the following video for instructions on linking an ORCID iD to your Editorial Manager account: https://www.youtube.com/watch?v=_xcclfuvtxQ.

Additional Editor Comments:

One reviewer suggested minor revision and one rejected the manuscript. Therefore, the academic editor suggests major revision.

Reviewers' comments:

Reviewer's Responses to Questions

**Comments to the Author**

1. Is the manuscript technically sound, and do the data support the conclusions?

Reviewer #1: No

Reviewer #2: Partly

2. Has the statistical analysis been performed appropriately and rigorously? 

Reviewer #1: N/A

Reviewer #2: N/A

3. Have the authors made all data underlying the findings in their manuscript fully available?

Reviewer #1: Yes

Reviewer #2: Yes

4. Is the manuscript presented in an intelligible fashion and written in standard English?

Reviewer #1: Yes

Reviewer #2: Yes

5. Review Comments to the Author

Reviewer #1: The manuscript contains a number of preliminary n=1 type experiments and there is no clear results. If the three samples of artificial or real saliva are replicates then the data is highly variable. If they are three different samples then it is not explained how they are different, There is no statistical analysis of the samples so it is unclear what are the significant results. The labelling on the graphs is very poor. The RGB/ spectral analysis of listerine when diluted with saliva also makes no sense While the application of microfluidics and impedance to the measurement of saliva is of interest this paper does not in anyway progress the field.

Reviewer #2: The manuscript submitted by Fatimah Ibrahim, Bojan Petrović et al. describes a microfluidic compact disc consisting of six unique chambers that run simultaneously from sample loading, holding, mixing and analysis used for the detection of the electrochemical property of different types of saliva and mouthwashes.

The manuscript is quite interesting but there are several issues that needs to be addressed by the authors before being accepted:

1. In the abstract please look at the phrase „On the other hand, electrochemical impedance property of artificial saliva, a commonly used drug for the treatment of xerostomia or dry mouth syndrome was also studied.” The drug is missing from this phrase

2. The microfluidic compact disc contained Al based electrodes. There are no issues related to the strong adsorption of the proteins from real saliva at the surface of those electrodes?

3. Are other similar approaches in the literature? Maybe it will be interesting to compare the findings from this manuscript withe the published results.

4. How the dick could be integrated into a point-of-care system? What parameters would be monitored by the end users of such a system?

5. The quality of figure 8 must be improved since the mentioned green color could not be observed.

6. PLOS authors have the option to publish the peer review history of their article (what does this mean?). If published, this will include your full peer review and any attached files.

Reviewer #1: No

Reviewer #2: No

---

## [Author Response · Author response to Decision Letter 0]

13 Dec 2022

Reviewer 1: "The manuscript contains a number of preliminary n=1 type experiments and there is no clear results. If the three samples of artificial or real saliva are replicates then the data is highly variable. If they are three different samples then it is not explained how they are different, There is no statistical analysis of the samples so it is unclear what are the significant results. The labelling on the graphs is very poor. The RGB/ spectral analysis of listerine when diluted with saliva also makes no sense While the application of microfluidics and impedance to the measurement of saliva is of interest this paper does not in anyway progress the field."

Response: 

" Thank you for the comment and suggestions. We have revised the manuscript performing additional experiments using real saliva samples from 3 different healthy volunteers instead of one reported in previous version. Methodology section was updated and error bars were included in Fig 4, Fig 5, Fig 6 and Fig 7. 

In short, three real saliva samples from 3 healthy subjects were obtained and used in the experiments. The microfluidic CD has six chambers and mouthwashes were mixed with artificial and real saliva in each single run (disc spin). Electrical impedance analysis was recorded for 3 times in each run. The entire experiment was repeated for 3 runs, resulting in n =27 for real saliva samples and n = 9 for artificial saliva (supplementary data is attached). The results indicate this novel microfluidics platform can reliably detect saliva conductivity enabling future clinical applications using rapid and low cost fabrication technique. We have performed statistical analysis to access the effect of mixing on electrical parameters on both real and artificial saliva samples.

RGB spectral analysis was demonstrated to show homogenous mixing in microfluidics CD (not the color analysis itself) as laminar flow in microchannels could result in non-homogenous mixture of fluids. Our microfluidics have incorporated specially designed channel to promote mixing of fluids.

We have added clarity in the manuscript for contribution of this research (marked in red font color). 

This paper demonstrate a new low cost fabrication technique for Compact Disc microfluidics using simple xurographic tools and integration of microfluidics with conductive electrodes for electrical impedance analysis. In addition, we demonstrate our microfluidic platform by performing fluidic mixing and impedance analysis on 6 analytes simultaneously. Our research demonstrated rapid, multiplex, low cost salivary analysis tool and open the way for further applications. First, by applying this system, it would be possible to conduct clinical studies with minimal amounts of salivary analytes in a faster and more efficient way. Then, at the level of individual point of care applications, with this study we have shown that the analysis of the electrical properties of saliva is feasible and simple, and each of the analyzed parameters is clinically relevant, both for healthy patients and for patients with oral diseases and disorders in the composition of saliva. 

In the conducted study, the saliva of healthy volunteers, without oral and general diseases was used, so the obtained values were in the physiological range, as expected. Future clinical studies will, we believe, confirm the effectiveness of this system in evaluating the electrical properties of saliva in patients with various disorders. However, even in healthy patients at low risk for developing oral diseases, it is possible to assess which type of the most commonly used mouthwash solutions lead to the most favorable changes in the composition of saliva, with the least chance of adverse effects.

Finally, in patients with reduced salivary secretion, it is possible to evaluate the effect of the artificial saliva that is used, as well as the combination of artificial saliva and antiseptic solutions that will reduce the symptoms, complaints and reduce the risk of oral diseases occurrence in patients with xerostomia."

Reviewer 2:

"The manuscript submitted by Fatimah Ibrahim, Bojan Petrović et al. describes a microfluidic compact disc consisting of six unique chambers that run simultaneously from sample loading, holding, mixing and analysis used for the detection of the electrochemical property of different types of saliva and mouthwashes.

The manuscript is quite interesting but there are several issues that needs to be addressed by the authors before being accepted:"

Response: "Thank you for the suggestions. We have improved our manuscript following comments and suggestion from the reviewer. "

1. In the abstract, please look at the phrase “On the other hand, electrochemical impedance property of artificial saliva, a commonly used drug for the treatment of xerostomia or dry mouth syndrome was also studied”. The drug is missing from this phrase. 

Response: 

We have revised the statement. 

“On the other hand, the electrochemical impedance property of artificial saliva, a commonly used moisturizing agent and lubricant for the treatment of xerostomia or dry mouth syndrome was also studied.”

"

2. The microfluidic compact disc contained Al based electrodes. There are no issues related to the strong adsorption of the proteins from real saliva at the surface of those electrodes?

Response:

In our study, we haven’t observed any adsorption of proteins. We rinsed the CD with DI water between each run with real saliva. No major changes were observed during repeated recordings.

3. Are other similar approaches in the literature? Maybe it will be interesting to compare the findings from this manuscript with the published results.

Response:

To the best of our knowledge, we haven’t found similar approach using microfluidics CD and impedance analysis of saliva in the literature. 

4. 1. How the disc could be integrated into a point-of-care system? What parameters would be monitored by the end users of such system?

Response: We have revised and included the suggestions in the manuscript. 

The possibilities of integrating this system into various clinical applications are numerous. First, by applying this system, it would be possible to conduct clinical studies with minimal amounts of salivary analytes in a faster and more efficient way. Then, at the level of individual point of care applications, with this study we have shown that the analysis of the electrical properties of saliva is feasible and simple, and each of the analyzed parameters is clinically relevant, both for healthy patients and for patients with oral diseases and disorders in the composition of saliva. 

In the conducted study, the saliva of healthy volunteers, without oral and general diseases was used, so the obtained values were in the physiological range, as expected. Future clinical studies will, we believe, confirm the effectiveness of this system in evaluating the electrical properties of saliva in patients with various disorders. However, even in healthy patients at low risk for developing oral diseases, it is possible to assess which type of the most commonly used mouthwash solutions lead to the most favorable changes in the composition of saliva, with the least chance of adverse effects.

Finally, in patients with reduced salivary secretion, it is possible to evaluate the effect of the artificial saliva that is used, as well as the combination of artificial saliva and antiseptic solutions that will reduce the symptoms, complaints and reduce the risk of oral diseases occurrence in patients with xerostomia.

5. The quality of figure 8 must be improved since the mentioned green color could not be observed. 

Response: 

Thank you. It was a mistake to have written ‘as shown in Figure 8’. Color is not clearly observable with naked eye. However, RGB analysis showed the difference in RGB values.

---

## [Editor Report · Decision Letter 1]

28 Dec 2022

A Novel Microfluidic Compact Disc to Investigate Electrochemical Property Changes Between Artificial and Real Salivary Samples Mixed with Mouthwashes Using Electrical Impedance Analysis

PONE-D-22-24397R1

Dear Dr. Ibrahim,

We’re pleased to inform you that your manuscript has been judged scientifically suitable for publication and will be formally accepted for publication once it meets all outstanding technical requirements.

Kind regards,

Ajaya Bhattarai

Academic Editor

PLOS ONE

Additional Editor Comments (optional):

The revised manuscript looks good.

Reviewers' comments:

<quillbot-extension-portal></quillbot-extension-portal>

---

## [Editor Report · Acceptance letter]

6 Feb 2023

PONE-D-22-24397R1 

A Novel Microfluidic Compact Disc to Investigate Electrochemical Property Changes Between Artificial and Real Salivary Samples Mixed with Mouthwashes Using Electrical Impedance Analysis 

Dear Dr. Ibrahim:

I'm pleased to inform you that your manuscript has been deemed suitable for publication in PLOS ONE. Congratulations! Your manuscript is now with our production department. 

Kind regards, 

on behalf of

Dr. Ajaya Bhattarai 

Academic Editor

PLOS ONE